# Conservative Evaluation of Offline Policy Learning

**Hager Radi Abdelwahed**                                        *radi@ualberta.ca*
*Department of Computing Science*
*University of Alberta, Edmonton, Canada*

**Josiah P. Hanna**                                              *jphanna@cs.wisc.edu*
*Computer Sciences Department*
*University of Wisconsin – Madison*

**Matthew E. Taylor**                                    *matthew.e.taylor@ualberta.ca*
*Department of Computing Science*
*Alberta Machine Intelligence Institute (Amii)*
*University of Alberta, Edmonton, Canada*

**Reviewed on OpenReview:** *https://openreview.net/forum?id=kLo4TKhOOP*

## Abstract

The world offers unprecedented amounts of data in real-world domains, from which we can develop successful decision-making systems. It is possible for reinforcement learning (RL) to learn control policies offline from such data but challenging to deploy an agent during learning in safety-critical domains. Offline RL learns from historical data without access to an environment. Therefore, we need a methodology for estimating how a newly-learned agent will perform when deployed in the real environment *before* actually deploying it. To achieve this, we propose a framework for conservative evaluation of offline policy learning (CEOPL). We focus on being conservative so that the probability that our agent performs below a baseline is approximately $\delta$, where $\delta$ specifies how much risk we are willing to accept. In our setting, we assume access to a data stream, split into a train-set to learn an offline policy, and a test-set to estimate a lower-bound on the offline policy using off-policy evaluation with bootstrap confidence intervals. A lower-bound estimate allows us to decide when to deploy our learned policy with minimal risk of overestimation. We demonstrate CEOPL on a range of tasks as well as real-world medical data.

## 1 Introduction

Suppose someone else is controlling a sequential decision making task for you. This could be a person trading stocks, a hand-coded controller for a chemical plant, or even a PID controller for temperature regulation. Offline reinforcement learning (RL) allows us to learn policies from historical data collected by some other controller. But when would one want to switch from the existing controller to the new policy? This decision may depend on the cost for continued data collection from the existing controller, your risk appetite, and your confidence in the performance of the policy you have learned. This paper takes a critical step towards the question: **how to (confidently) select the right time to deploy when learning offline, if we do not have access to the environment nor the policy generating the data?**

Offline reinforcement learning is a way to train off-policy algorithms using existing data. It presents a great opportunity for learning data-driven policies without environment interaction. In safety-critical applications such as in healthcare or autonomous driving, there is a large amount of data that we can use to learn RL policies and hence use for decision making (Gottesman et al., 2019). Learning with offline data is challenging because of the distribution mismatch between the data collected by the behavior policy that collected the data, and the offline agent, which learns from data (Levine et al., 2020). What is even more challenging is evaluating offline agents in the offline RL setting if we assume no access to the environment; in some

domains, we cannot execute our learned policy until it is good enough because it can be costly or dangerous if it performs worse than the current policy or controller. This limitation raises the possibility of using off-policy evaluation (OPE) methods, where we can estimate the value of a policy using trajectories from another policy, to predict what the performance of the offline agent is at any point of time during training. We further investigate if it is better to use OPE with confidence intervals to control the risk of overestimating the policy's performance, which is referred to as conservative off-policy evaluation ($COPE$).

We present a framework for conservative evaluation for offline policy learning (CEOPL). CEOPL tackles the setup where we combine offline policy learning with conservative off-policy evaluation; we learn a policy from pre-existing data and evaluate it to provide a confidence lower-bound estimate on its return. The goal is to learn a target policy purely from data and evaluate its performance while learning so that we can tell when it is ready for deployment. In conservative off-policy evaluation ($COPE$), we ensure safety by using bootstrap confidence intervals to provide a lower-bound on the policy value estimates using a reasonable amount of test data. $COPE$ provides an approximate solution to high confidence off-policy evaluation (HCOPE); HCOPE estimates a lower bound on the policy's value that is correct with a set level of confidence, but requires a substantial amount of data to achieve a tight lower bound.

Our target setup is summarized in Figure 1: we have some source of data, from which we request data samples. At each step, data is split into training and testing. After each offline learning step, we test the policy using conservative evaluation. A policy passes the safety test if a lower-bound on the policy's value is better than the value of the data distribution. We continue the process of training/testing for a few iterations until the testing shows the policy can outperform the existing controller with a confidence level $\delta$. We dynamically receive samples, continuously perform RL updates, and continuously monitor a confidence interval on the changing policy until it reaches a sufficient level for deployment. We hypothesize that our conservative off-policy evaluation ($COPE$) is preferred over off-policy evaluation (OPE) because OPE may be prone to overestimation, which is problematic for safety-critical problems. CEOPL acts as a workflow for offline learning and evaluation in safety-critical domains. Contributions of this article are summarized as follows:

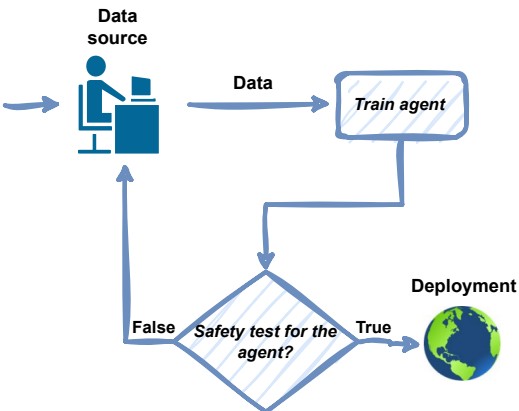

Figure 1: **Setup for continual safety-evaluation of offline RL**: this figure illustrates the loop of interaction between an offline agent and data while the agent is trained, evaluated and deployed if it passes the safety test.

- CEOPL, a framework combining offline RL with conservative off-policy evaluation for both discrete and continuous control tasks.

- Studying the feasibility of offline evaluation without access to an environment nor the behavior policy.

- Exploring the effect of offline learning on OPE methods given constantly-improving target policy (as opposed to a fixed policy, as previously studied in the literature).

- Demonstrating CEOPL on real-world medical data.

## 2 Related Work

In this section, we provide a review of the current literature of offline reinforcement learning and off-policy evaluation. We also discuss the gap in research our work is trying to fill.

**Offline Reinforcement Learning**   Offline reinforcement learning refers to RL algorithms learning from a fixed dataset $\mathcal{D}$ without environment interaction or further data collection. Such data is the transition tuples

$\langle s_t, a_t, r_t, s_{t+1}, \gamma_{t+1} \rangle$ resulting from historical agent-environment interaction, which will eventually compose a trajectory $H$ of size $L$. We learn $\pi_\theta$, a new target policy that is not necessarily similar to the behavior policy $\pi_b$. In this work, we assume we have access to a dataset $\mathcal{D} = \{H_1, H_2, ..., H_n\}$ as a set of $n$ trajectories generated by an unknown behavior policy $\pi_b$.

For offline RL, a survey paper (Levine et al., 2020) categorizes offline RL algorithms into model-free methods and model-based methods. Our paper only considers model-free approaches for offline RL, where model-free methods are sub-categorized into policy constraint methods that constrain the learned policy to be close to the behavior policy and uncertainty-based methods that attempt to estimate the epistemic uncertainty of Q-values to reduce distributional shift. In policy gradient methods, a policy constraint can be enforced directly on the actor update to keep the new policy as similar as possible to the behavior policy while learning. This constraint can be implicit KL-divergence, which does not model the behavior policy; for example, see advantage-weighted regression (AWR) (Peng et al., 2019) or advantage weighted actor-critic (AWAC) (Nair et al., 2020). A constraint can also be an explicit divergence constraint that require an estimation for the behavior policy as done in the way off-policy (WOP) algorithm (Jaques et al., 2019). In practice, policy constraint methods so far seem to outperform pure uncertainty-based methods, as shown with the batch constrained deep Q-learning paper (Fujimoto et al., 2019b). In bootstrapping error accumulation reduction (BEAR) (Kumar et al., 2019), authors identify bootstrapping error as the source of instability of offline RL methods; they propose doing distribution-constrained backups via maximum-mean discrepancy (MMD) (Kumar et al., 2019). In behavior-regularized offline reinforcement learning (Wu et al., 2019), a general framework, behavior constrained actor-critic (BRAC), is introduced to cover different ways of regularization to offline policies, whether as a value penalty or a policy penalty. Non-constrained methods include traditional Q-learning (Watkins & Dayan, 1992), double DQN (Hasselt et al., 2016), and soft actor critic (SAC) (Haarnoja et al., 2018), which are not always successful in the fully offline setup. Behavioral cloning (BC) (Bain & Sammut, 1999), as a main method for imitation learning, is another way for learning offline policies from historical data; BC proved to perform well compared to offline RL methods under medium-quality and expert data (Kumar et al., 2019).

**Off-policy Evaluation**  Off-policy evaluation (OPE) methods evaluate a target policy $\pi_\theta$ using data generated by a behavior policy $\pi_b$. When the off-policy estimate is guaranteed with some confidence that its performance is not worse than the behavior policy $\pi_b$, it is referred to as high confidence off-policy policy evaluation (HCOPE) (Thomas et al., 2015a). An empirical study of OPE methods (Voloshin et al., 2019) discussed the applicability of both methods and presented method selection guidelines depending on the environment parameters and the mismatch between $\pi_\theta$ and $\pi_b$. This study categorized OPE methods into importance sampling (IS) methods, direct methods, and hybrid methods that combine aspects from both IS and direct methods. Importance Sampling (IS), or Inverse Propensity Scoring, as it is known in statistics, is one of the most widely used methods for off-policy evaluation where rewards are re-weighted by the ratio between $\pi_\theta$ and $\pi_b$ (Precup et al., 2000). This weighting results in a consistent and unbiased off-policy estimator. Later versions of importance sampling provide lower-variance estimates such as weighted importance sampling (WIS) (Mahmood et al., 2014), per-decision importance sampling (PDIS), and per-decision weighted importance sampling (PDWIS) (Precup et al., 2000). To avoid bias due to using the off-policy state distribution and the high variance of importance weights, Dice-style methods (Nachum et al., 2019a) (Nachum et al., 2019b) (Zhang et al., 2020) (Yang et al., 2020) estimate the marginal importance ratio between state density distributions instead. Secondly, direct methods rely on regression techniques to directly estimate the value function of the policy without access to the behavior policy. This category includes model-based methods where the transition dynamics and reward are estimated from historical data via a model. Then, the off-policy value is computed with Monte-Carlo policy evaluation in the model (Hanna et al., 2017). To achieve conservative OPE, MB-bootstrap (Hanna et al., 2017) uses model-based off-policy estimator with bootstrapping to construct confidence intervals for OPE estimates. A recent work introduced HAMBO (Rothfuss et al., 2023), which uses an uncertainty-aware learned model of the transition dynamics to obtain tight lower bounds of policies. Another direct method is fitted Q-evaluation (FQE) (Le et al., 2019), a model-free approach that acts as the policy evaluation counter-part to batch Q-learning or fitted Q-iteration (FQI) (Riedmiller, 2005). The last category contains the hybrid methods that combine different features from IS and direct methods, which mainly involves doubly-robust methods (Jiang & Li, 2016) that

use a direct model to lower IS variance. A few other methods improve upon this approach, such as weighted doubly-robust (WDR) (Thomas & Brunskill, 2016), and MAGIC (Thomas & Brunskill, 2016).

**Gap in The Literature**  In the offline RL literature, there is no standard way to evaluate a policy while learning offline; most of the literature still evaluates the performance of offline algorithms in the environment or in a simulator, by running a policy online. A workflow for offline RL (Kumar et al., 2021) proposed a set of metrics to indicate how the algorithm can be further improved, such as detecting over-fitting and under-fitting, providing no clear workflow for estimating the policy performance. At the intersection of offline RL and OPE, there has been recent work combining these two areas. One paper discussed the applicability of FQE (Le et al., 2019) to test the performance of offline policies under different hyper-parameters and select the best performance (Paine et al., 2020). Active offline policy selection (Konyushova et al., 2021) uses OPE to warm-start the evaluation process and choose which policy to evaluate online when given a limited budget of online interactions. In benchmarks for deep off-policy evaluation (Fu et al., 2021), authors evaluate offline policies learned on continuous control tasks using various OPE methods ranging from importance sampling, model-based methods, and doubly-robust estimators. They show how challenging OPE can be as an evaluation method (Fu et al., 2021).

Even considering the previous work combining offline RL with OPE (Paine et al., 2020) (Fu et al., 2021), none discussed the feasibility of evaluating offline RL agents as we learn $\pi_\theta$, and how much we can trust OPE as an approach for testing. The setup we are studying is quite different from the current literature because previous work for off-policy evaluation assumed a behavior policy $\pi_b$ and a target policy $\pi_\theta$, where both policies are fixed and may be related. For example, in a study for high-confidence OPE methods (Thomas et al., 2015a), the target policy is initialised as a subset of the behavior policy such that they are close to each other. In another study for safe improvement (Thomas et al., 2015b), the Daedulus algorithm learns a safe target policy as a continuous improvement over the behavior policy. This is quite relevant to our work; however, it is not purely offline learning because their algorithm uses data from an older version of the policy it currently improves to perform both the improvement step and safety tests. With the doubly-robust estimator, results are presented with different versions of $\pi_\theta$ such that $\pi_\theta$ is always a mixture of $\pi_b$ and $\pi_\theta$ with different degrees to ensure their relevance (Jiang & Li, 2016).

Recent work on offline RL focuses on continuous control in near-complex domains and environments (Levine et al., 2020). As detailed above, OPE literature focuses on much simpler domains in discrete and continuous control (Voloshin et al., 2019). None of the previous work discussed the feasibility of using conservative OPE to evaluate offline RL policies. We believe the setup we are tackling is under-studied in the literature where the target policy is not necessarily similar to the data collection policy, which is the case for offline RL.

## 3  Methodology

In this section, we introduce our framework CEOPL in detail with the methods used for both learning and evaluation of an offline policy. CEOPL couples offline RL algorithms with conservative off-policy evaluation as a feasible evaluation method for offline agents in safety-critical domains. Conservative evaluation is defined by any off-policy evaluation method which seeks a high-probability lower bound on the policy's estimate. In CEOPL, we sample data from a buffer of data dynamically at each iteration, perform policy updates, and continuously

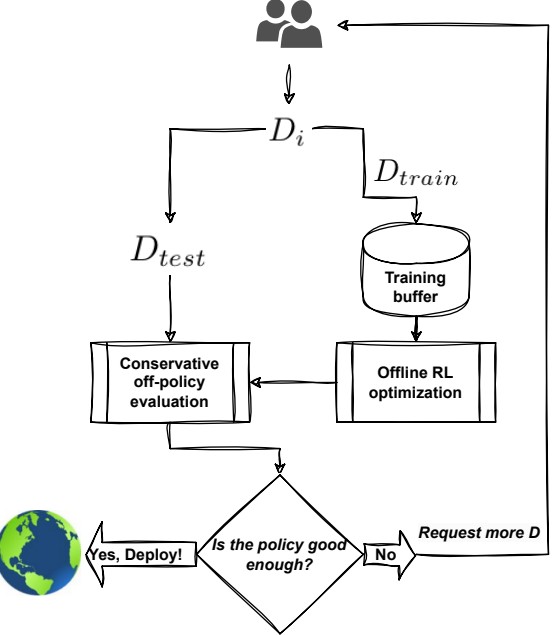

Figure 2: **Proposed workflow for CEOPL**: this figure shows the methodology to follow, when given a source of data, to both learn a policy offline and estimate its value confidently.

monitor its performance with a confidence interval
until it reaches a reasonable performance and is ready to be deployed. The policy learning offline does not interact with the environment unless it passes the safety test. Moreover, we assume no control or access to the behavior policy $\pi_b$ producing the data, but we assume that we can request more data as long as our offline policy does not pass the safety test. However, the same methodology applies if we just have access to a static dataset of trajectories (which we demonstrate in Section 6). We have $k$ iterations where, in each iteration $i$, dataset of trajectories $D_i$ of size $n$ is requested from the main data source. $D_i$ is split between $D_{train}$ to train our offline policy $\pi_{\theta_i}$ and $D_{test}$ to perform the safety test using conservative off-policy evaluation methods. $D_{train}$ is accumulated over iterations in the training data buffer while a new set of $D_{test}$ is used each evaluation iteration for the confidence interval to hold. CEOPL terminates when the learned policy is ready to be deployed with appropriate confidence. Instantiating CEOPL requires selecting an offline policy learning method and an off-policy evaluation method for computing off-policy estimates. We study various methods for offline policy learning and evaluation as our methodology. Figure 2 shows an overview of CEOPL framework based on a given data source, where the interaction between an agent and the data source is summarized as follows:

1. An agent receives a set of data $D_i$ without access to the behavior policy.

2. We optimize a policy $\pi_{\theta_i}$ offline using $D_{train}$ with an offline RL algorithm (Section 3.1).

3. We perform a safety test using $D_{test}$ with a conservative off-policy evaluation method (Section 3.2).

4. Our new policy is either ready to be deployed or we go back to step 1.

---

**Algorithm 1** Conservative Evaluation for Offline Policy Learning (CEOPL) Framework

---

**Input**: initial $\pi_\theta$, dataset $D$ of $m$ trajectories, confidence level $\delta \in [0, 1]$, number of trajectories needed every iteration $n$, number of bootstrap estimates $B$, offline learning method $\Phi$, and an OPE method $\Psi$
**Output**: $\pi_\theta$, $\hat{v}_\delta(\pi_\theta)$: $1 - \delta$ lower-bound on $\hat{v}(\pi_\theta)$

1: Let $i = 0$
2: **while** $\hat{v}_\delta(\pi_\theta) \leq \hat{v}(\pi_b)$ **do**
3:     Request a set of trajectories $D_i$ with size $n$
4:     Split $D_i$ into $D_{train_i}$ and $D_{test}$
5:     $D_{train} = D_{train} \cup D_{train_i}$
6:     Optimize policy $\pi_{\theta_i} = \Phi(D_{train})$
7:     Evaluate policy $\hat{v}_\delta(\pi_\theta) = \text{BCI}(\pi_\theta, D_{test}, \delta, B, \Psi)$
8: **end while**
9: **return** $\pi_\theta$, $\hat{v}_\delta(\pi_\theta)$

---

We refer to each offline policy learning method with $\Phi$ and each off-policy evaluation method with $\Psi$. CEOPL is further explained in Algorithm 1. The inputs are a dataset of trajectories $D$, a confidence level $\delta$ appropriate for the problem in hand, offline policy learning method $\Phi$, and an off-policy evaluation method $\Psi$ to combine with bootstrap confidence intervals, shown in Algorithm 2 in Appendix A.3. The output will be a policy trained offline $\pi_\theta$ and a confidence lower-bound estimate of its return $\hat{v}_\delta(\pi_\theta)$. Since the stopping condition in Algorithm 1 is not always satisfied if no evaluation method can detect an improved policy, we set a maximum number of iterations for training and evaluation in our experiments.

## 3.1 Methodology: Offline Reinforcement Learning

Is it possible to learn a good policy offline given data and no environment interaction? Earlier work (Agarwal et al., 2020) tried to answer this question, and showed how capable RL algorithms are in the offline setting, since they were able to learn a successful DQN agent totally offline when provided with diverse data. However, a benchmark on batch RL (Fujimoto et al., 2019a) showed how RL without correction can fail in the offline setting given ordinary data.

Table 1: **Categories for offline learning**: this table shows different categories for offline learning methods for both discrete and continuous control, with examples of algorithms used in our experiments.

|  | **Discrete Control** | **Continuous Control** |
|---|---|---|
| **Off-policy** | Double DQN (Hasselt et al., 2016) | SAC (Haarnoja et al., 2018) |
| **Imitation Learning** | Behavioral Cloning (Bain & Sammut, 1999) | Behavioral Cloning |
| **Offline** | BCQ (Fujimoto et al., 2019b) | BCQ & BEAR (Kumar et al., 2019) |

In CEOPL, we investigate different techniques for learning from offline data, ranging from normal off-policy RL algorithms (off-policy), imitation learning techniques (imitation learning), and policy constraint methods specific for offline learning (offline). Imitation learning is a form of supervised learning and is good at learning from expert data or demonstrations. Policy constraint methods are designed to mitigate the challenges for normal off-policy to learn offline, such as bootstrapping error (Levine et al., 2020). This categorization is inspired by the offline RL tutorial (Levine et al., 2020) and is shown in Table 1. Details about each algorithm are further explained in Appendix A.1. Given the data samples used from a data source, we aim to learn a policy totally offline without environment interaction or access to the policy generating the data. This offline policy is expected to be at least as good as the behavior policy in case of imitation learning and better in the case of other algorithms, after a few iterations.

### 3.2 Methodology: Conservative Off-policy Evaluation

Conservative off-policy evaluation extends OPE methods to lower-bounding the performance of the target policy, $\pi_\theta$, when combined with bootstrapping confidence intervals.

**Off-Policy Evaluation**

While a policy is being improved offline, we are interested in evaluating such a policy. Off-policy evaluation is a promising technique to evaluate a policy that is continuously learning given access to fixed data. An off-policy estimator is a method for computing an estimate $\hat{v}(\pi_\theta)$ for the true value of the target policy $v(\pi_\theta)$ using trajectories $D$ collected while following another policy $\pi_b$, which is what OPE methods do. When evaluating, we use new samples for testing each time to avoid the multi-comparison problem[2]. This sampling is also to ensure that we do not over-fit our OPE estimates or tune training parameters to reduce the estimate error. Table 2 shows the three categories of OPE methods, with examples under each category. The methods in bold are the ones we use for evaluation. For instance, we use Weighted Importance Sampling (WIS) (Mahmood et al., 2014), Direct Model-based (MB) (Hanna et al., 2017), and Weighted Doubly-Robust (WDR) Thomas & Brunskill (2016) estimators as shown. More details are further shown in Appendix A.2.

Table 2: **Different OPE methods categorized**: This table shows the three main categories of OPE methods, importance sampling, direct methods and hybrid methods with examples with the methods used in our experimentation.

| **Importance Sampling** | **WIS** (Mahmood et al., 2014) |
|---|---|
| **Direct** | **Model-based** (Hanna et al., 2017) |
| **Hybrid** | **Weighted doubly-robust**(Thomas & Brunskill, 2016) |

**Bootstrapping**

To conservatively evaluate in CEOPL, each OPE method is combined with bootstrapping to approximate a confidence lower bound of $\hat{v}_\delta(\pi_\theta)$ on $\hat{v}(\pi_\theta)$ such that $\hat{v}_\delta(\pi_\theta) \leq \hat{v}(\pi_\theta)$ with probability at least $1 - \delta$. In principle, bootstrapping can be replaced by any other method that provides tight lower bounds. Consider a data sample $D$ of $n$ random variables $H_j$ for $j = 1, 2, ..., n$ where we can sample $H_j$ from some *i.i.d.*

---

[2]The problem occurs when conducting multiple statistical tests simultaneously with reusing data, so the confidence bound does not strictly hold anymore.

distribution of data. From the sample of data $D$, we can compute a sample estimate $\hat{x}$ of a parameter $x$ such that $\hat{x} = f(D)$ where $f$ is any function that estimates $x$. Given a dataset $D$, we create $B$ resamples with replacement, where $B$ is the number of bootstrap resamples, and compute an estimate for $x$, $\hat{x}$, on each of these resamples. Bootstrapping (Efron, 1979) allows us to estimate the distribution of $\hat{x}$ with confidence bounds. The estimates computed with different resamples will determine the $1 - \delta$ confidence interval. In our setup, the parameter of interest $x$ is the expected return of a policy $v(\pi_\theta)$.

With a confidence level $\delta \in [0, 1]$ and $B$ resamples of the dataset of trajectories $D$, we use bootstrapping to approximate a lower-bound confidence interval of $v(\pi_\theta)$ on $v(\pi_\theta)$ such that $v_\delta(\pi_\theta) \leq v(\pi_\theta)$ with probability at least $1 - \delta$. This bootstrap method is referred to as the percentile bootstrap for confidence intervals (Carpenter & Bithell, 2000). After obtaining $B$ resamples of the dataset, we compute $\hat{v}(\pi_\theta)$ with each of these resamples. Then, all the off-policy estimates $\hat{v}(\pi_\theta)$ are sorted ascendingly and the index $\delta \times B$ is chosen to calculate $\hat{v}_\delta(\pi_\theta)$. When bootstrapping is combined with off-policy evaluation methods, we can estimate the distribution of $v(\pi_\theta)$, and use it to estimate $\hat{v}_\delta(\pi_\theta)$ with probability at least $1 - \delta$. We refer to the algorithm used as bootstrap confidence intervals (BCI) and the pseudo-code is detailed in Algorithm 2 in Appendix A.3. Although bootstrap confidence intervals provide approximate high-confidence estimates, they provide a practical approach with small amounts of data compared to exact concentration inequalities that may be too loose to use.

## 4 Experiments

This section discusses the experiments and the results of CEOPL on simulated setups in discrete and continuous control tasks. In our experiments, two simulated environments (MountainCar-v0 and Pendulum-v0) were used to demonstrate CEOPL in discrete and continuous control, respectively. This choice shows how the complexity of the environment, state and action dimensions, and horizon are affecting offline evaluation. To mimic the setup where there is a source of trajectories without access to the environment itself, we simulate a source of data in each environment as a partially-trained policy that collects data of medium quality. Such data resulting from a policy that is not optimal nor random resembles real-world data and leaves room for improvement when training offline RL policies. For each environment, a policy is optimized offline using three different offline learning methods and evaluated simultaneously using three different off-policy evaluation methods. Following Algorithm 1, in each iteration, we get $m$ trajectories from our data source which we split between training and testing. We use more data for testing than training since CEOPL requires large amounts of data to estimate tight bounds. For each improvement method, we load the train split into the training buffer and do $k$ training epochs where $k$ varies among offline RL algorithms. Since we exceptionally have access to the true environment in these simulated domains, we compute the true value of the offline policy by testing it in the actual environment and use it for comparison. For bootstrapping, we use $\delta = 0.05$ to get a 95% confidence lower-bound using $B = 2000$ bootstrap estimates, as recommended by practitioners (Efron, 1979). In weighted doubly-robust with bootstrapping 3.2, we use a value of $B = 224$ for the number of bootstrap estimates to avoid heavy computation; this can still get us a good approximation as suggested by MAGIC (Thomas & Brunskill, 2016). Since we have no access to $\pi_b$, we estimate $\hat{\pi}_b$ given the test data (Hanna et al., 2021) with a behavior cloning model that is improved gradually with more test data.

**Results on MountainCar-v0**: For each iteration, 300 trajectories are sampled, where 20 trajectories are used into the training buffer and 280 trajectories are used for evaluation. This totals to 3000 trajectories over the 10 iterations. Each evaluation iteration includes performing $k$ training epochs to the offline policy; a training epoch is a single optimization step of the policy over a batch of data. $k$ is set to be $2^9$ for BCQ and Double DQN, while BC does $2^6$ policy updates in one iteration (since BC was faster to optimize). Figure 3 shows how different offline policy improvement methods perform given medium-quality data with conservative evaluation, indicating which method can tell when $\hat{v}_\delta(\pi_\theta) > \hat{v}(\pi_b)$. While the x-axis shows the number of evaluation iterations, the y-axis shows the value estimate of a policy across different iterations. Behavior cloning, as an offline policy improvement method, can only perform as well as $\pi_b$. Double DQN and BCQ were able to outperform $\pi_b$. The true value of a target policy is calculated as the average return when running the policy in the actual environment (not possible in practice) for 1000 episodes. All reported results are an average of 40 runs, while the shaded area shows the standard error. The value of the behavioral policy, $\hat{v}(\pi_b)$, is the sum of undiscounted rewards of all trajectories in the data set.

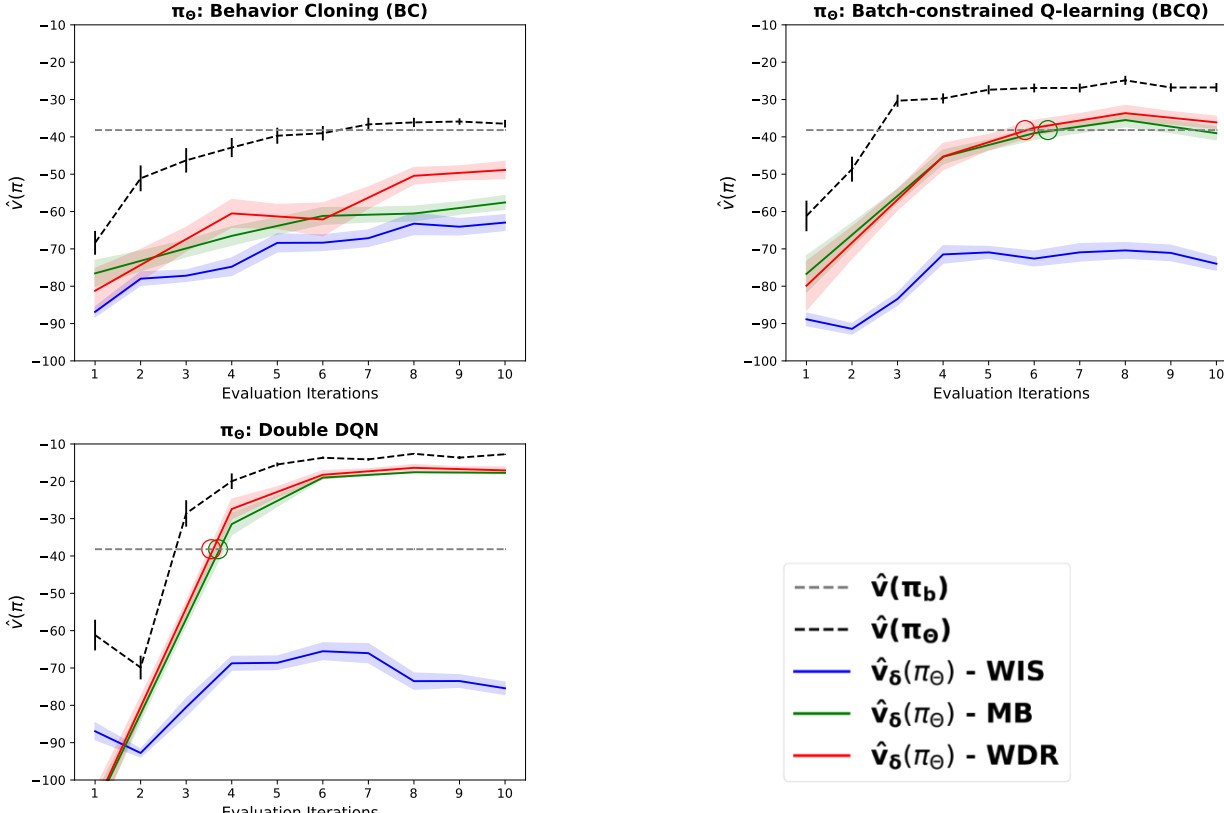

Figure 3: **CEOPL on MountainCar-v0**: this figure shows the performance of the target policy as it is training offline over iterations (x-axis). Weighted importance sampling (WIS) failed to detect that the offline policy outperforms $\pi_b$. The model-based estimator (MB) and weighted doubly-robust (WDR) were able to determine when an offline agent outperforms $\pi_b$. While the y-axis shows the estimated policy value $\hat{v}(\pi)$, we show the empirical estimate $\hat{v}(\pi_b)$ for the behavioral policy (given the dataset), and the conservative off-policy estimate $\hat{v}_\delta(\pi_\theta)$ across different estimators.

For conservative evaluation of each offline agent, weighted importance sampling (WIS) with bootstrapping failed to detect that the offline policy outperforms $\pi_b$ for all learning methods. The model-based estimator (MB) and weighted doubly-robust with bootstrapping (WDR) have much less error with respect to the true value of a policy and were able to infer when an offline agent outperforms $\pi_b$ and hence is ready to be deployed. For instance, with 95% confidence, in the case of offline improvement with Double DQN, we were able to tell that our new target policy is better than the behavior policy at iteration 4 using two different estimators (MB and WDR).

**Results on Pendulum-v0**: For each iteration, 500 trajectories are sampled, where 100 trajectories goes into the training buffer and 400 trajectories for evaluation. This totals 5000 trajectories over the 10 iterations. Figure 4 shows how different offline policy improvement methods perform given medium-quality data with conservative evaluation, indicating which method can tell when $\hat{v}_\delta(\pi_\theta) > \hat{v}(\pi_b)$. Behavioral cloning as an offline policy improvement method can only perform as well as the data by $\pi_b$, while BEAR and BCQ were able to outperform $\pi_b$. We did not report results on SAC (Haarnoja et al., 2018) since it failed to learn a policy better than the behavior policy. In MountainCar-v0, the true value of a target policy is calculated as the average return when running the policy in the actual environment (which is not possible in practice) for 1000 episodes. All reported results are an average of 40 runs, while the shaded area shows the standard error. $\hat{v}(\pi_b)$ is the sum of undiscounted rewards of the data set, which represents the value of the behavioral policy. Similar to the results on MountainCar-v0, WIS with bootstrapping failed to detect that the offline policy outperforms $\pi_b$. The MB estimator achieve a very tight lower-bound with the true value of a policy

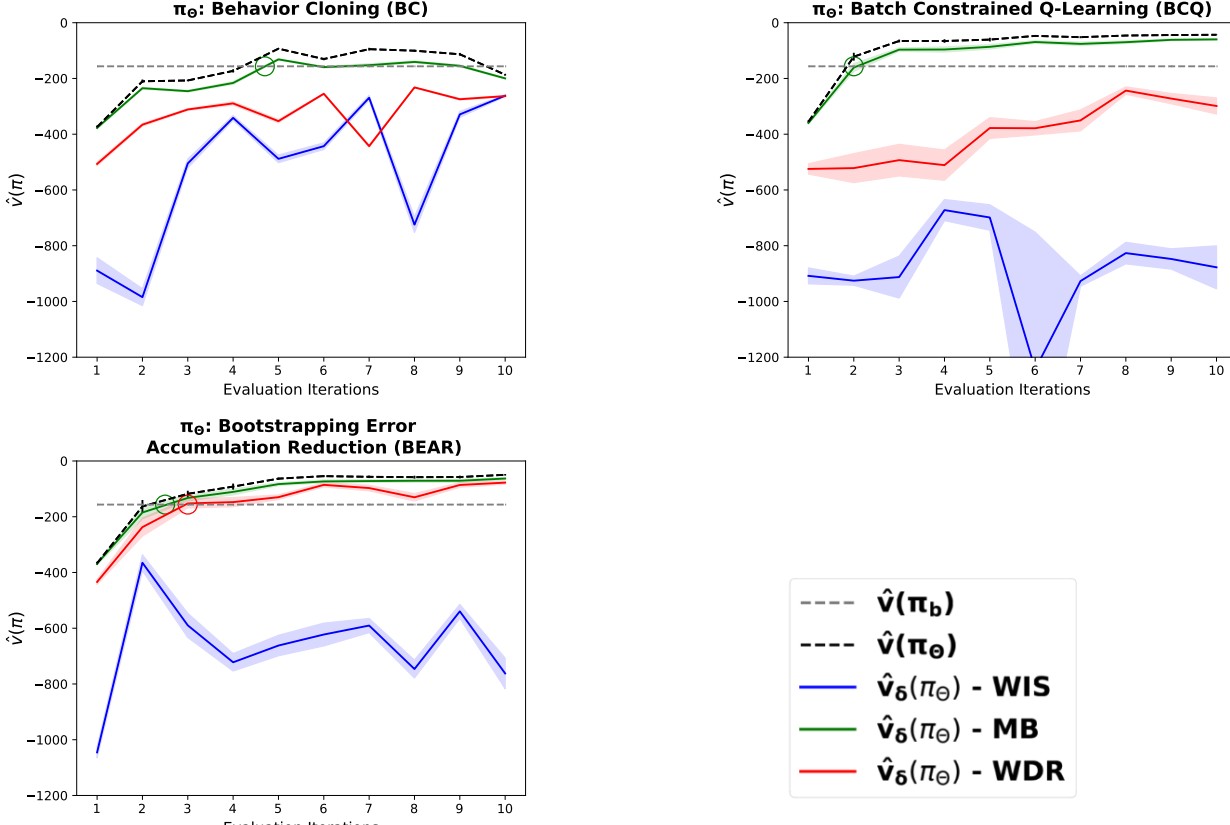

Figure 4: **CEOPL on Pendulum-v0**: this figure shows the performance of the target policy as it is training offline over iterations (x-axis). Weighted importance sampling (WIS) failed to detect that the offline policy outperforms $\pi_b$. The model-based estimator (MB) and weighted doubly-robust (WDR) were able to determine when an offline agent outperforms $\pi_b$. While the y-axis shows the estimated policy value $\hat{v}(\pi)$, we show the empirical estimate $\hat{v}(\pi_b)$ for the behavioral policy (given the dataset), and the conservative off-policy estimate $\hat{v}_\delta(\pi_\theta)$ across different estimators.

and were able to identify when an offline agent outperforms $\pi_b$ and therefore is ready to be deployed. WDR estimates are also affected by the improvement methods as in the case with WIS.

## 5 Analysis & discussion

### 5.1 Overestimation of OPE vs. Conservative OPE:

It is important to question whether conservative evaluation in our proposed framework is worth the added complexity and computation of bootstrapping over the vanilla off-policy evaluation (OPE). Although both OPE and conservative OPE can provide value estimates of an offline target policy, conservative OPE lower-bound estimates underestimate the value of the offline policy which serves our purpose of ensuring safety. Moreover, most existing OPE estimators are prone to overestimating the value of the policy Thomas et al. (2015a). Empirical results on simulated environments showed how OPE estimates can overestimate the true value of the offline policy, as shown in Figure 5. Even if not all OPE estimators are overestimating the true value of the offline policy in Figure 5, there are no guarantees on the performance of the reported OPE estimators. On the other hand, lower bounds in CEOPL are guaranteed to only overestimate the true value of the offline policy within the allowable 5% error rate for a 95% confidence, as shown in Figure 7 in Appendix B.2. Bootstrapping has strong guarantees as the size of test data goes to $\infty$, but it lacks guarantees for finite samples (Hanna et al., 2017); this is because it assumes that the bootstrap distribution is representative of the true distribution of the statistic of interest. As a result of this assumption, bootstrapping is considered

| | *Off-policy* (Double DQN) | *Imitation Learning* (BC) | *Offline* (BCQ) |
|---|---|---|---|
| WIS | $49.032 \pm 0.57$ | $\mathbf{28.742} \pm 1.22$ | $43.66 \pm 0.84$ |
| MB | $\mathbf{7.85} \pm 0.72$ | $22.716 \pm 1.09$ | $13.645 \pm 0.65$ |
| WDR | $\mathbf{6.18} \pm 1.45$ | $17.642 \pm 2.79$ | $12.81 \pm 2.42$ |

Table 3: **Mean error between *CEOPL* estimates and true value of $\pi_\theta$ on MountainCar-v0**: this table shows the mean error of conservative OPE estimates, $\hat{v_\delta}(\pi_\theta)$, over all training iterations and multiple runs given each offline improvement method. Lowest mean errors for each method are shown in bold., and standard error over 40 runs is reported.

| | *Imitation Learning* (BC) | *Offline* (BCQ) | *Offline* (BEAR) |
|---|---|---|---|
| WIS | $\mathbf{268.5} \pm 28.93$ | $602.49 \pm 78.05$ | $432.22 \pm 47.5$ |
| MB | $33.48 \pm 0.77$ | $25.02 \pm 2.5$ | $\mathbf{19.0} \pm 5.1$ |
| WDR | $161.1 \pm 13.47$ | $307.11 \pm 37.3$ | $\mathbf{54.75} \pm 10.43$ |

Table 4: **Mean error between *CEOPL* estimates and true value of $\pi_\theta$ on Pendulum-v0**: this table shows the mean error of conservative OPE estimates, $\hat{v_\delta}(\pi_\theta)$, over all training iterations and multiple runs given each offline learning method. Lowest mean errors for each method are shown in bold, and standard error over 40 runs is reported. SAC is not reported as it failed to learn a good policy.

semi-safe because the assumption it makes may be false. Prior work Kostrikov & Nachum (2020) identified conditions, such as sufficient data size and sufficient coverage, under which statistical bootstrapping is guaranteed to yield correct confidence intervals. Other work (Morrison et al., 2007) shows that bootstrapping is still safe enough for high-risk predictions with a known record of producing accurate confidence intervals. Therefore, it is better to rely on CEOPL for conservative evaluation in safety-critical applications.

## 5.2   What affects CEOPL?

If we are to apply CEOPL to any problem, OPE methods and offline RL methods are different factors affecting the performance. To investigate this, we do further analysis and report total variation (TV) distance, a measure of similarity between two probability distributions; this is to check how the learning method of an offline policy is affecting TV distance between $\hat{\pi}_b$ and $\pi_\theta$ and hence affecting the off-policy estimates. Results are shown in Figure 8 in Appendix B.3, where the total variation distance between the offline policy $\pi_\theta$ and the estimated behavior policy $\hat{\pi}_b$ across the different offline learning methods is reported. For discrete control, behavioral cloning (imitation learning) can achieve a much lower distance than other improvement methods (batch-constrained Q-learning and double DQN) whether they are constrained or non-constrained. This is because behavioral cloning forces its target policy to be close to the behavior policy while other methods do not. The same applies to continuous control in Pendulum-v0; behavior cloning achieves the lowest error between $\hat{\pi}_b$ and $\pi_\theta$. This insight on divergence explains why weighted importance sampling achieves the lowest error between the estimate and the true value in the case of behavioral cloning, as shown in Tables 3 and 4; the error grows for other methods that do not constrain the policy to be close to the data distribution. This analysis suggests that the choice of a OPE method is dependant on the offline method chosen to optimize the policy. Other factors affecting the performance include the horizon of the trajectory and the environment dynamics, which we discuss further in Appendix B.3.

**Takeaways**: the performance of conservative off-policy evaluation, and OPE in general can be affected by the following factors: a) divergence between $\hat{\pi}_b$ and $\pi_\theta$ as discussed earlier in Section 5.1, b) the horizon of the trajectory, c) the environment dynamics. The horizon of the trajectory affects importance sampling such that it suffers from high variance with longer horizons (Liu et al., 2020). The environment dynamics also affect direct and hybrid methods. The MB estimator performs well when it is possible to model the environment dynamics such as the case for both MountainCar-v0 and Pendulum-v0 environments. The WDR estimator also performs well because q-values can get more accurate when it is easy to learn a model of the environment. However, the WDR estimator is still affected by the divergence between $\hat{\pi}_b$ and $\pi_\theta$ as

in the case for BC and BCQ algorithms. WDR estimates get worse as the divergence increases. As a result, we can conclude that

- When it is possible to model the environment dynamics, we should rely on estimators that do not take $\pi_b$ into account (e.g., direct methods or hybrid methods) so that value estimates are less affected by the divergence between $\pi_\theta$ and $\hat{\pi}_b$ (as the case for WIS).

- In short-horizon environments, we should rely on importance sampling methods when the offline policy algorithm is constrained to be similar to $\hat{\pi}_b$.

Based on our empirical analysis in subsections 5.1 and 5.2, we conclude that conservative off-policy estimators are considered a safe evaluation method for offline learning with enough confidence that minimizes the risk of overestimating the true performance of an offline policy. In contrast, off-policy evaluation suffers from overestimation to the true values so it makes sense to rely on conservative evaluation in CEOPL instead, for domains where safety is essential. Further, our ability to accurately detect an improved policy with CEOPL is a function of different factors: the offline learning algorithm, the environment dynamics and the trajectories horizon.

## 6 Case Study: Real world data

In this section, we demonstrate our proposed framework CEOPL on real-world medical data to see how our framework can perform.

### 6.1 Problem: Sepsis Treatment

Problems in healthcare can be considered a form of sequential decision making, such as clinical physicians making decisions about the best next step to take in care (Ghassemi et al., 2019). Current advances in AI can provide personalized care that is equal to or better than humans by finding an optimal decision making policy given clinical data (Ghassemi et al., 2019). Sepsis, a severe infection with organ failure, is a leading cause of mortality in intensive care units (ICUs) in hospitals (Sakr et al., 2018). There are continuous efforts in research to understand and cure sepsis and, hence, increase the survival rates for ICU patients. RL for sepsis treatment is preferred over supervised learning because the ground truth of good treatment strategy is un-

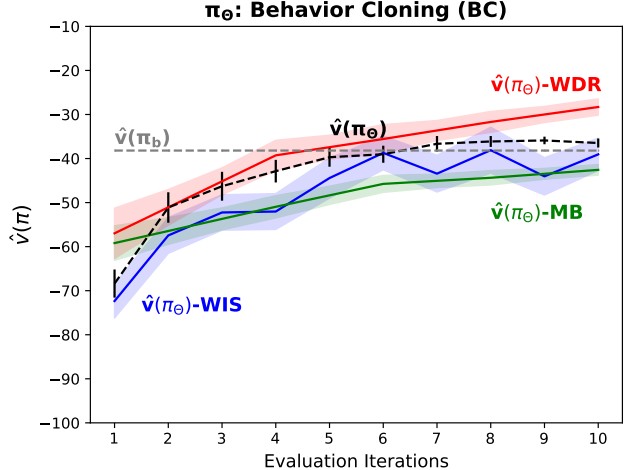

Figure 5: **Results of vanilla off-policy evaluation on MountainCar-v0**: figure shows results of evaluating a policy, optimized offline with behavioral cloning, with OPE (without bootstrapping). It shows that OPE overestimates the true value of $\pi_\theta$.

clear in the current medical literature (Marik, 2015). Sepsis data is a form of offline data, from which we can learn an offline policy given no access to an environment. The challenge lies in evaluating such policies while learning to determine when it is good enough to be deployed. We use the well-known MIMIC III data (Johnson et al., 2016) for sepsis treatment. Current literature in RL for health (Raghu et al., 2017a;b) finds it challenging to evaluate policies learned from such data given no agent-environment interaction.

For this problem, given no further data collection, it is better to rely on offline RL methods (rather than online RL) to learn a policy from a fixed dataset. A recent work (Killian et al., 2020) explores different representation learning techniques to learn a good policy offline with discrete BCQ (Fujimoto et al., 2019a). To evaluate their policies, they rely on Weighted Importance Sampling (Mahmood et al., 2014) to compare the different representation learning techniques. The problem at hand is widely researched but, only this previous work (Killian et al., 2020) relied on offline learning with off-policy evaluation. While relying on importance sampling for evaluation is good when comparing different approaches, we believe it is essential

to perform evaluation with confidence for this critical application to avoid the risk of overestimation. We build upon their work to apply our proposed training-evaluation framework.

## 6.2 Experimental Design

We would like to perform conservative evaluation for offline policy learning and see if we can trust our RL policy to make decisions in the real-world. To apply CEOPL, there are a few design decisions to make first. These decisions include: how much data to use for training/testing, bootstrapping size, confidence level, and choice of algorithm to optimize a policy offline. Given that the available data is fixed, we split the dataset once in the beginning into a train set, used to train the offline policy, and a test set, used for evaluating the policy performance during training. Note that this fixed batch setup is different from what we followed in Section 4 where we assumed a growing dataset. With stratified sampling (Killian et al., 2020), we use a 70/30 train/test split, which maintains the same proportions of each terminal outcome (survival or mortality). Data description and experimental details are shown in Appendix C.1.

Unlike simulated environments in Section 4, we do not have a way in this setup to evaluate how a policy performs except for using off-policy estimators. With CEOPL, we use weighted importance sampling (WIS), model-based estimator (MB), and weighted doubly-robust estimator (WDR) with bootstrapping to provide lower-bound estimates $\hat{v_\delta}(\pi_\theta)$ for $\pi_\theta$. We also report $\hat{v}(\pi_\theta)$ using the same estimators without bootstrapping, as the OPE estimate. Data is split between training and evaluation. Training data is used to optimize a BCQ agent while testing data is used for evaluation with conservative OPE methods.[3] With evaluation, we check if the agent is good enough to deploy in the real-world, and if not, we need to continue improving our offline agent before deploying. For this problem, we hypothesize that importance sampling (Precup et al., 2000) will provide good estimates given the short horizon and the offline policy constraint method used for improvement, unlike the MB estimator which will find transition dynamics for this problem challenging.

## 6.3 Results

For sepsis data, there is no ground-truth to compare against, as opposed to the simulated environments we presented in Section 4. While learning our offline policy, we would like to see how well it performs over time and when it outperforms the behavior policy so we can deploy it into the real world.

Figure 6 shows the results of applying CEOPL to train and evaluate an offline RL policy, as an average of 10 runs with the standard error indicated. The value of the behavior policy, $v(\pi_b)$, is the average return of the data in hand. With WIS, the conservative estimate $\hat{v_\delta}(\pi_\theta)$ and the mean estimate $\hat{v}(\pi_\theta)$ show the improvement of the policy over training iterations. The mean OPE estimate of WIS slightly outperforms the behavior policy but this can be an overestimate of the true value. We focus on our conservative estimate, which did not show that the target policy outperforms the behavior policy. The conservative estimates by the model-based estimator (MB) failed to detect the improvement of the policy over time. This has to do with failing to model the complex dynamics of the dataset; as a result, it failed to estimate the true value of the target policy. In a model-based estimator, we model all the environment dynamics, the next state, reward, and terminal state. In MountainCar-v0, we only modeled the next state since the reward and terminal state can be computed deterministically from the current state. In Pendulum-v0, the environment always terminates at the maximum horizon, so we only modeled the next state and reward. With sepsis data, it was hard enough to model the next state because the state dimension is large; it is also hard to model the terminal state indicating when a patient dies and whether they survive or not.

With weighted doubly-robust estimator, values of $\hat{v_\delta}(\pi_\theta)$ and $\hat{v}(\pi_\theta)$ can show the improvement of the offline policy as it is training, until both estimates outperform $\hat{v_\delta}(\pi_\theta)$. WDR combines per-decision weighted importance sampling with an approximate model; the approximate model only fits q-values so it does not suffer from the modeling complexity as the MB estimator. However, we cannot rely on the mean estimates $\hat{v}(\pi_\theta)$ as they may overestimate. For real-world sepsis data, we suggest that relying on OPE only is not enough for such critical application, and it is better to rely on CEOPL for conservative lower-bound estimates

---

[3]For this case study, we face the multi-testing problem in this experiment as we used the same data split for each evaluation iteration; this is because the size of the test data is too small to split among iterations.

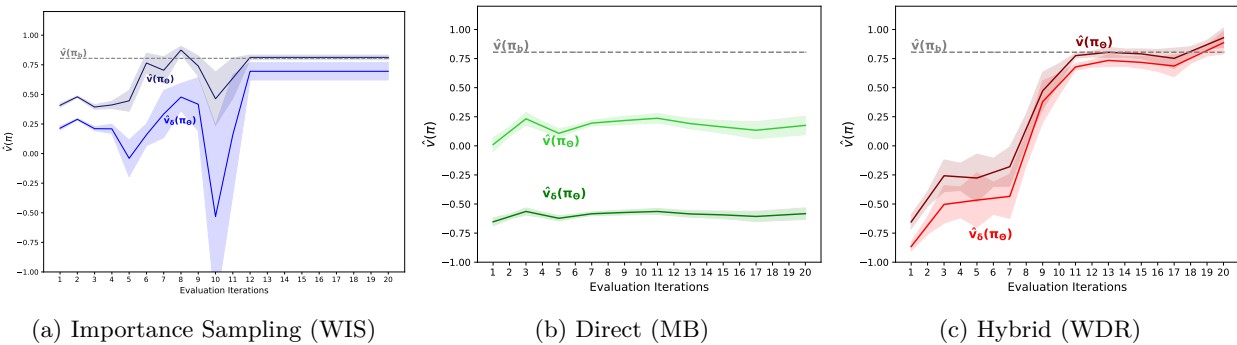

(a) Importance Sampling (WIS)  (b) Direct (MB)  (c) Hybrid (WDR)

Figure 6: **Results of CEOPL on sepsis data**: this figure shows results of evaluating the performance of the offline policy. We show both the conservative off-policy estimate $\hat{v}_\delta(\pi_\theta)$, and mean OPE estimate $\hat{v}(\pi_\theta)$.

because it will be less likely to overestimate. As for the choice of an evaluation method, WIS can be reliable for this problem given the short horizon but WDR can provide better estimates as a hybrid method that improves upon importance sampling. It is challenging for the MB estimator to detect an improved policy given the complex environment dynamics.

## 7   Conclusion

This paper proposed a framework for evaluating offline RL methods in a conservative manner (CEOPL), combining OPE with bootstrap confidence intervals. First, we studied the feasibility of different categories of conservative off-policy estimators and how they are affected by offline learning algorithms. We suggest that conservative evaluation is better for safety-critical applications since off-policy evaluation may suffer from overestimation. We tested CEOPL on real-world medical data for sepsis treatment. While dynamically receiving data, we optimize offline RL agents and run safety tests to estimate a lower-bound on the value of the offline policy and control the risk of overestimating its true value. This is essential for safety-critical applications to be able to tell when it is safe enough to deploy a new policy. In conclusion, conservative evaluation with CEOPL proved to be a reliable evaluation method for offline agents that do not have access to a simulated environment. When combined with bootstrapping, direct and hybrid OPE methods can be trusted for evaluating offline agents. The horizon of the trajectory and the environment dynamics affect the choice for the off-policy estimator to rely on. In future work, we plan to tackle learning offline from multiple data sources with different qualities and explore how the quality of the data affects offline learning and conservative evaluation. In addition, we can explore other OPE methods such fitted Q-evaluation (Le et al., 2019) to eliminate the challenges of direct model-based evaluation, and explore recent approaches for conservative evaluation such as HAMBO (Rothfuss et al., 2023) which are more reliable in continuous state-action spaces.

### Broader Impact Statement

We do not foresee any potential negative impacts of this research that users need to be aware of. Additionally, our framework provides a safe way to perform evaluation of offline RL methods. We have showcased a real-world application of our proposed framework in healthcare, where offline RL can be utilized safely, and hence have a positive impact.

### Acknowledgments

Part of this work has taken place in the Intelligent Robot Learning (IRL) Lab at the University of Alberta, which is supported in part by research grants from the Alberta Machine Intelligence Institute (Amii); a Canada CIFAR AI Chair, Amii; Digital Research Alliance of Canada; Huawei; Mitacs; and NSERC.

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

# A    Appendix: Methodology

This section provides further details of the methods used in CEOPL.

## A.1    Offline Reinforcement Learning

As mentioned in Section 3.1, we categorized learning methods into three categories: normal off-policy RL algorithms (off-policy), imitation learning techniques (imitation learning), and policy constraint methods specific for offline learning (offline). For continuous control, we describe the approaches in the three categories. For the off-policy category, we used soft actor-critic (SAC) (Haarnoja et al., 2018); this means a SAC agent is optimized offline using a fixed buffer of data without interaction with the environment or control over the buffer. For the imitation learning category, we study Behavioral Cloning (BC) (Bain & Sammut, 1999); BC is a supervised learning algorithm where the model learns how to predict actions from the current state.

In the offline category, we study batch-constrained Q-learning (BCQ) (Fujimoto et al., 2019b) and bootstrapping error accumulation reduction (BEAR) (Kumar et al., 2019). Batch-constrained Q-learning (Fujimoto et al., 2019b) argue that off-policy algorithms often fail in the offline setting due to the extrapolation error since unseen state-action pairs will have unrealistic values (Fujimoto et al., 2019b). This is a result of the mismatch between the state-action visitation of the current policy and the state-action pairs in the offline dataset. To solve this, the policy should induce a similar state-action visitation to the offline data. BCQ (Fujimoto et al., 2019b) proposed the use of a state-conditioned generative model to produce only likely actions to the current offline data. Then, this generative model, effectively a variational auto-encoder (Kingma & Welling, 2013), is combined with a network which aims to optimally perturb the generated actions in a small range. When this is combined with the Q-network, the network will only select the highest valued actions similar to the data in the batch (Fujimoto et al., 2019b).

BEAR (Kumar et al., 2019) argues that the source of instability for off-policy algorithms learning offline is the bootstrapping error. This error results from bootstrapping with actions that lie outside of the training data distribution, and it accumulates via the bellman backup operator (Kumar et al., 2019). BEAR is built on top of any actor-critic algorithm, such as SAC (Haarnoja et al., 2018), by modifying the policy improvement step to use a distribution-constrained backup. To apply the constraint, the sampled version of maximum-mean discrepancy is used between the unknown behavior policy $\pi_b$ and the current actor $\pi_\theta$ to constrain the distribution of the actor to the support of the behavior policy (Kumar et al., 2019). Hence, BEAR allows the actor to maximize the Q-function while being constrained to remain in the valid support space of the behavior policy defined by the data samples (Kumar et al., 2019).

For discrete control, we use Double DQN (Hasselt et al., 2016) as a normal off-policy RL algorithm; similar to SAC, Double DQN is used to learn solely from offline data. For the imitation learning category, we use behavioral cloning (BC) (Watkins & Dayan, 1992) in a supervised learning manner with cross-entropy loss. In the offline category, we experiment with the discrete version of BCQ (Fujimoto et al., 2019a). Discrete BCQ is much simpler than its continuous version, since it trains Q-learning with a constrained argmax operator. This only allows actions in the backup with probability, given by the generative model, above some threshold $\tau$. The generative model is a behavioral cloning network trained in standard supervised learning with a cross-entropy loss (Fujimoto et al., 2019a). BCQ is also based on Double DQN, and we use the threshold to be $\tau = 0.3$ as reported in their original paper. If the threshold $\tau$ is 1, the algorithm returns an imitator of all the actions in the dataset, while a threshold $\tau = 0$ returns the Q-learning objective.

## A.2    Off-Policy Evaluation

**Importance Sampling**   (Precup et al., 2000) is a method for handling mismatch between distributions and hence presented as a consistent and unbiased off-policy estimator. For a trajectory $H \sim \pi_b$ of length $L$, as $H = s_1, a_1, r_1, .., s_L, a_L, r_L$, we can define the importance sampling up to time $t$ for policy $\pi_\theta$ as follows:

$$IS(\pi_\theta, \pi_b, D) = \sum_{i=1}^{m} \rho_L^H R^i, \quad \rho_t^H := \prod_{j=0}^{t} \frac{\pi_\theta(A_j|S_j)}{\pi_b(A_j|S_j)} \tag{1}$$

By re-weighting the returns, we can tell how likely each reward is under $\pi_\theta$ versus $\pi_b$. However, IS often assumes a known $\pi_b$, which is not always the case for off-policy policy evaluation. Since we assume no access to $\pi_b$ in our setting, we estimate a behavior policy $\hat{\pi}_b$ from data with a behavior cloning model (Hanna et al., 2021). To come up with a high confidence estimate, we first compute the importance-weighted returns then use bootstrapping, described in Section 3.2, to get the lower-bound estimate. In our experiments, we use weighted importance sampling (WIS) (Mahmood et al., 2014) with bootstrapping as a representation for importance sampling family of estimators.

$$WIS(\pi_\theta, D, \pi_b) = \sum_{i=1}^{m} \frac{\rho_{L-1}^i}{\sum_{j=1}^{m} \rho_{L-1}^j} g(H_i) \tag{2}$$

**Direct Model-Based Estimator** is another off-policy estimator that falls under direct methods. The model-based off-policy estimator (MB) computes $\hat{v}(\pi_\theta)$ by building a model using all the available trajectories $D$ to build a model $\hat{M} = (S, A, \hat{P}, r, \gamma, \hat{d}_0)$ where $\hat{P}$ and $\hat{d}_0$ are estimated with trajectories sampled from $\pi_b$. Then, MB will compute $\hat{v}(\pi_\theta)$ as the average return of trajectories simulated in the estimated model $\hat{M}$ while following $\pi_\theta$. Despite having lower variance than IS methods, MB is a biased estimator for: 1) we lack data for particular state-action pairs, so we assume their transition probabilities, 2) we assume that the model class includes the true model for the transitions. As a result, as $n \to \infty$, the model estimates may converge to a value different from the true $v(\pi_\theta)$. This effect is due to its dependency on the modeling assumptions we make whether we assume a linear or a non-linear model.

Therefore, the model-based estimator is combined with bootstrapping, as discussed in Section 3.2, for conservative off-policy evaluation. To clarify, as mentioned in Section A.1, we rely mainly on model-free policy optimization algorithms for learning a policy offline. However, we use model-based estimation to evaluate a policy, which is independent from the policy optimization part.

**Weighted Doubly-Robust Estimator** (Thomas & Brunskill, 2016) is a hybrid method for off-policy estimation, presented as an extension to the doubly-robust (DR) method (Jiang & Li, 2016). DR is an unbiased estimator of $v(\pi_\theta)$ that uses an approximate model of the MDP to reduce the variance of importance sampling (Jiang & Li, 2016). Although biased, WDR is based on per-decision weighted importance sampling (PDWIS) and improves upon the DR method as it balances the bias-variance trade-off. In addition, the approximate model value functions act as a control variate for PDWIS.

$$PDWIS(\pi_\theta, D, \pi_b) = \sum_{i=1}^{m} \sum_{t=0}^{L-1} \frac{\rho_t^i}{\sum_{j=1}^{m} \rho_t^j} \gamma^t R_t^i \tag{3}$$

$$WDR(\pi_\theta, D, \pi_b) = PDWIS(\pi_\theta, D, \pi_b) -$$
$$\sum_{i=1}^{n} \sum_{t=0}^{L-1} \gamma^t (w_t^i \hat{q}_{\pi_\theta}(S_t^i, A_t^i) - w_{t-1}^i \hat{v}_{pi_\theta}(S_t^i)) \tag{4}$$

Similar to direct model-based methods, we use bootstrapping with WDR to provide a confidence lower-bound estimate on $\hat{v}(\pi_\theta)$. WDR with bootstrapping is guaranteed to converge to the correct estimate as $n$ increases, given the statistical consistency of PDWIS (Hanna et al., 2017). For the approximate model, a single model is estimated with the available trajectories $D$, then used to compute the value functions of WDR for each bootstrap data. We choose the weighted doubly robust estimator to represent OPE hybrid methods in our study.

### A.3 Bootstrapping

The pseudo-code of Bootstrap Confidence Intervals (BCI) is shown in Algorithm 2.

---

**Algorithm 2** Bootstrap Confidence Intervals: BCI

---

**Input**: a target policy $\pi_\theta$, dataset $D_{test}$ of $m$ trajectories, confidence level $\delta \in [0, 1]$, number of bootstrap estimates $B$, an estimate of the behavior policy $\hat{\pi}_b$, and off-policy estimator $\Psi$

**Output**: $\hat{v_\delta}(\pi_\theta)$: $1 - \delta$ lower-bound on $\hat{v}(\pi_\theta)$

1: **for** $j \in [1, B]$ **do**
2: $\quad \tilde{D}_j = H_1^j, .., H_m^j$ where $H_i^j$ is sampled uniformly
3: $\quad \hat{v} = \Psi(\pi_\theta, \tilde{D}_j, \hat{\pi}_b)$
4: **end for**
5: $\text{sort}(\hat{v}_j | j \in [1, B])$ // ascending
6: $l = \lfloor \delta B \rfloor$
7: **return** $\hat{v}_l$

---

# B    Appendix: Empirical Results

This section provides further details on the experimental setup and further analysis of the empirical results on the simulated environments.

## B.1    Simulated Environments

**MountainCar-v0**: we use discrete MountainCar (Sutton & Barto, 2018) with a continuous state (velocity and position) and 3 possible discrete actions. At each time-step, the reward is $-1$, except for the terminal state when it is 0. However, we used the modified version of Mountain-Car as described here (Thomas, 2015). The horizon is shortened by holding an action $a_t$ constant for 4 updates of the environment state. We also change the start state such that an episode starts with a random position in the range of (-1.2, 0.6) and random velocity in the range of (-0.07, 0.07) (Jiang & Li, 2016; Thomas, 2015). The data collector we use for this environment is an online actor-critic (Sutton & Barto, 2018) agent that is partially trained with added 30% randomization when generating data. This means when collecting the offline data, an agent takes a random action 30% of the time instead of following the online policy, to include exploratory transitions.

**Pendulum-v0**: inverted pendulum swing-up problem is a classic problem in the control literature with one continuous action (Brockman et al., 2016). In this version of the problem, the pendulum starts in a random position, and the goal is to swing it up to stay upright. We also shorten the horizon following the same procedure as done in MountainCar (Thomas, 2015) by holding an action constant for 4 updates of the environment state. This limits the horizon of the environment to 50 instead of 200. The data source we use for this environment is a soft actor-critic (Haarnoja et al., 2018) agent that is trained partially, with added 30% randomization when generating the dataset. This means when collecting the offline data, an agent takes a random action 30% of the time instead of following the online policy to resemble real-world data.

## B.2    Overestimation of CEOPL

Figure 7 shows the overestimation of lower bounds compared to the true value of the offline policy; it is guaranteed to remain within the allowable $\delta = 5\%$ error rate for a 95% confidence. We would like to point out that the safety guarantee for the confidence bound specified is per-iteration (Thomas, 2015). If we want to consider this guarantee over multiple iterations, the probability $1 - \delta$ decreases for the early iterations as we perform more tests. As a result, we should only consider this guarantee for the last iteration of our loop, such that the error rate in the last iteration is at most $\delta$.

## B.3    Further Analysis

To better understand how evaluation is affected by the offline learning method, multiple measures can show the similarity between two probability distributions (as a policy is a distribution over actions). One simple measure is the total variation (TV) distance. TV distance measures the difference between action probabilities taken under two policies given the data set in hand. TV would be the sum of differences in

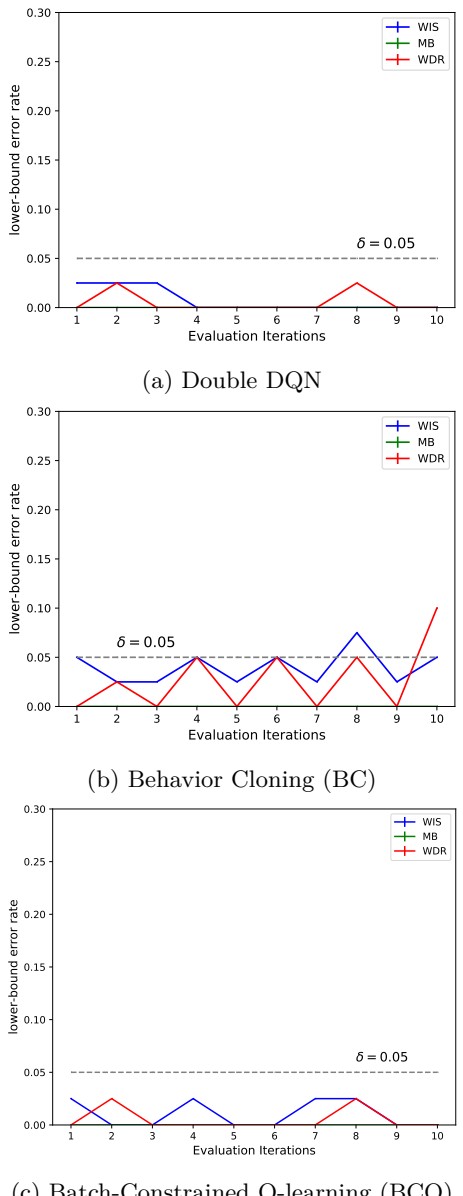

(a) Double DQN

(b) Behavior Cloning (BC)

(c) Batch-Constrained Q-learning (BCQ)

Figure 7: **The empirical error rate on MountainCar-v0 with CEOPL**: This figure shows the empirical error rate on MountainCar given different evaluation methods. The lower bound is computed $z$ times for each method ($z = 40$ for Mountain Car) and we count how many times the lower bound is above the true $\hat{v}(\pi_\theta)$ computed in the true environment. Given 200 trajectories only for evaluation, all methods correctly approximate the allowable 5% error rate for a 95% confidence lower bound. However, there are two instances of WIS and WDR that slightly exceed 5%, which can be mitigated with more data.

probabilities between the behavior policy $\pi_b$ and the target policy $\pi_\theta$ for each state-action pair in the test dataset. Since $\pi_b$ is not known, we estimate $\hat{\pi}_b$ given the test data (Hanna et al., 2021). TV distance is correlated to KL-divergence, showing how two policies are different from each other. As discussed in Section 5.2 and shown in Figure 8, in MountainCar-v0 (discrete control), behavior cloning (BC) learns a policy that is very similar to $\hat{\pi}_b$. Discrete BCQ and double DQN do not limit the distance with the behavior policy, and their TV distance is quite large as opposed to BC. For Pendulum-v0 (continuous control), behavior cloning

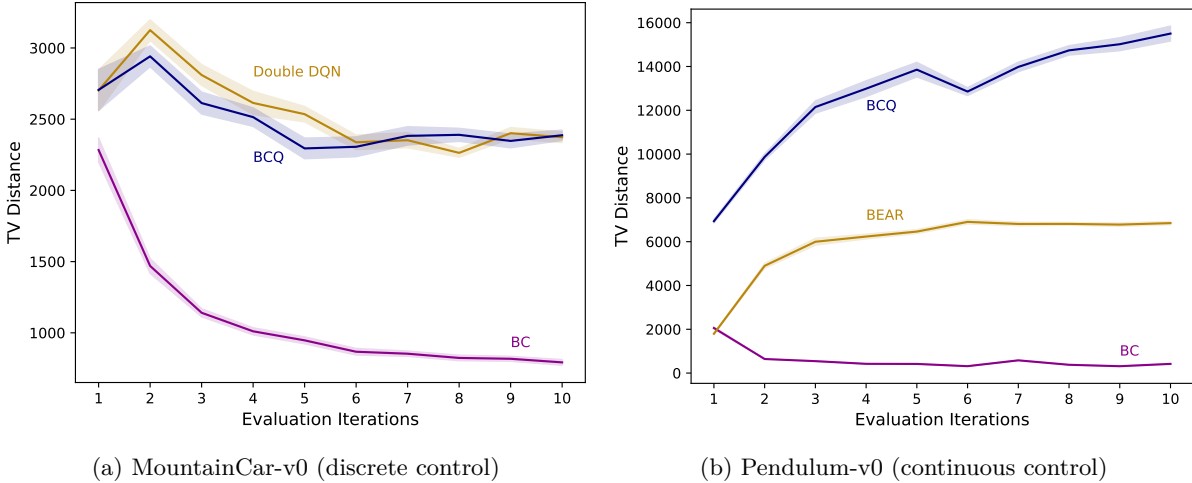

(a) MountainCar-v0 (discrete control)    (b) Pendulum-v0 (continuous control)

Figure 8: **Divergence between $\hat{\pi}_b$ & $\pi_\theta$ over training iterations**: this figure shows the total variation (TV) distance between $\hat{\pi}_b$ & $\pi_\theta$ on MountainCar-v0 & Pendulum-v0 over all training iterations. For both discrete (a) and continuous control (b), BC achieves the lowest distance since it tries to mimic the data distribution. In (a), Double DQN and BCQ policies achieve similar distance, but higher compared to BC. (b), BEAR achieves a higher distance than BC, but much less than BCQ; this is because BEAR enforces explicit divergence minimization between $\hat{\pi}_b$ and $\pi_\theta$.

(BC) also learns a policy that is very similar to $\hat{\pi}_b$ and achieves the lowest TV distance. BEAR limits the divergence compared to BCQ since BEAR's objective is constrained to be close to $\pi_b$, as shown in Figure 8.

## C   Appendix: Real-world Case Study

This section provides further experimental details and analysis of the experiments on sepsis data.

### C.1   Experimental Details

To build a decision-making policy for the treatment of septic patients, we use data from the Medical Information Mart for Intensive Care (MIMIC-III) dataset (v1.4) (Johnson et al., 2016). We follow previous relevant work (Komorowski et al., 2018) to extract and preprocess vital and lab measurements and build a cohort of 19418 patients; this cohort has an observed mortality rate just above 9% (determined by death within 48h of the final observation). Data extraction, preprocessing, and splitting follow previous work (Killian et al., 2020).

A dataset $D$ is a collection of trajectories; a trajectory $H$ of a non-fixed horizon $L$ refers to a single patient that ends by the survival or death of a patient, with a maximum horizon $L$ of 19. A transition is composed of current observation $O_t$, the action taken by the clinician $a_t$ to move to the next observation $O_{t+1}$ receiving reward of $r_t$ until a patient survives or dies. We treat the problem as a markov decision process (MDP). Data consists of a continuous state space of a dimension 38 and a discrete action space of 25 actions. Observed actions are the administration of fluids or vasopressors that can be given, categorized by volume, and put into 5 discrete bins per action type resulting in 25 actions (Killian et al., 2020). The data has sparse reward such that a trajectory has a terminal reward of +1 if a patient survived, -1 if a patient died, and 0 otherwise. The observations used as our state space are a combination of time-varying continuous features (33), such as heart rate, glucose, etc., and demographic features (5), such as gender, age, weight, etc. (Killian et al., 2020).

For the bootstrapping size, we use $B = 2000$ as in the experiments with simulated data. We also use a confidence level $\delta = 0.05$. We optimize a policy offline using batch-constrained Q-learning (BCQ) for discrete

control (Fujimoto et al., 2019a), coupled with an auto-encoder for better state representation learning. Another important decision is to decide how often we will run our conservative evaluation methods because of its computational cost. For this, we train the policy for $200k$ epochs, where we evaluate the policy every $10k$ epochs. To estimate the behavior policy $\pi_b$, which is referred to as the clinician's policy, we rely on behavior cloning (Hanna et al., 2021) as done in other experiments on simulated environments.

