# OpenReview forum: "Conservative Evaluation of Offline Policy Learning"
_TMLR — Accepted by TMLR_

### Review · Reviewer_Ksxj · 2024-04-26

**Summary Of Contributions:**

The authors propose a framework for conservative evaluation of offline policy learning (CEOPL). They focus on being conservative so that the probability that the agent performs below a baseline is approximately $\delta$, where $\delta$ specifies how much risk we are willing to accept. They assume access to a data stream, split into a training set to learn an offline policy, and a testing set to estimate a lower bound on the offline policy using off-policy evaluation with bootstrap confidence intervals.

**Audience:**

Yes

**Claims And Evidence:**

Yes

**Requested Changes:**

1. While the authors claim that a lower bound estimate of policy evaluation is established. There is no theory to support that $\hat v$ produced by Algorithm 1 is a lower bound with probability $1-\delta$.
2. I can see the benefit of conservative approach in policy learning. But for policy evaluation, if one $\pi$ is overestimated, would every $\pi$ be overestimated? So just be a constant shift?

**Strengths And Weaknesses:**

Strengths: The paper is well-written.
Weaknesses: The paper lacks theoretical results.

---

> ### Author Response · Authors · 2024-05-16
> **Authors' response to reviewer Ksxj**
>
> We would like to thank reviewer Kxsj for their timely review. We reply to the requested changes below:
>
> > While the authors claim that a lower bound estimate of policy evaluation is established. There is no theory to support ...
>
> We would like to point to bootstrapping’s wide use in statistics despite no non-asymptomatic guarantees in general. Although bootstrapping has no theoretical guarantees, it provides a practical alternative to guaranteed bounds that are too loose to use.  Previous work in healthcare [1] shows that bootstrapping is still safe enough for high-risk predictions with a known record of producing accurate confidence intervals. Prior work in OPE, like [2] and [3], have previously established the use of bootstrapping for off-policy evaluation confidence intervals. [4] identified conditions (sufficient data size and sufficient coverage) under which statistical bootstrapping is guaranteed to yield correct confidence intervals. Such conditions are not much of an issue in our finite domains. We have added further discussion about this in Section 5.1. We reiterate what reviewer pbpT mentioned: “The application of bootstrapping to get confidence intervals is a quick but reasonable and effective way to get a lower-bound estimate on the value, which was our motivation.”
>
> > I can see the benefit of conservative approach in policy learning. But for policy evaluation, if one $\pi$ is overestimated, would every $\pi$ be overestimated? So just be a constant shift?
>
> For policy evaluation, It is not true that if one $\pi$ overestimates that every $\pi$ is overestimated. It is not a constant shift because off-policy evaluation depends on the offline policy and its distance from the behavioural policy (which we measure by total deviation distance), so the shift depends on policy changes as training progresses. Our framework proposed that It is better to underestimate the value of the offline policy $\pi$, and provides empirical guarantees that it can underestimate the value of the policy with probability $1- \delta$.
>
>
> [1] Prediction of coronary heart disease risk using a genetic risk score: the Atherosclerosis Risk in Communities Study, https://pubmed.ncbi.nlm.nih.gov/17443022/
>
> [2] High Confidence Policy Improvement https://proceedings.mlr.press/v37/thomas15.pdf
>
> [3] Bootstrapping with Models: Confidence Intervals for Off-Policy Evaluation https://arxiv.org/pdf/1606.06126
>
> [4] Statistical Bootstrapping for Uncertainty Estimation in Off-Policy Evaluation, https://arxiv.org/abs/2007.13609

---

> > ### Comment · Reviewer_Ksxj · 2024-05-19
> > **reviewer's comment**
> >
> > From a theoretical point of view, I don't see the claim that it is better to underestimate the value of the offline policy $\pi$. I am wondering if the authors can clarify.

---

> > > ### Author Response · Authors · 2024-05-19
> > > **Authors' reply to reviewer Ksxj's comment**
> > >
> > > Thanks for your comment. It is a well-known issue in the OPE literature that most OPE estimators suffer from overestimation. According to Thomas et. al 2015 [1], most existing OPE estimators suffer from high variance and are prone to overestimating the performance of the policy. In [2], they explain how OPE methods make predictions under strong distribution shifts, and as a result, their estimates suffer from high variance and are prone to overestimate the performance of the policy. Both papers are cited in our work. As a result, our work relies on empirical results and analysis to show OPE overestimation bias. We further suggest that it is safer to underestimate policy values instead, to mitigate the overestimation issues, and provide conservative estimates via bootstrapping.
> > >
> > > [1] High-confidence off-policy evaluation, https://people.cs.umass.edu/~pthomas/papers/Thomas2015.pdf
> > >
> > > [2] Hallucinated Adversarial Control for Conservative Offline Policy Evaluation, https://arxiv.org/pdf/2303.01076

---

### Review · Reviewer_pbpT · 2024-05-03

**Summary Of Contributions:**

The authors discuss off-policy evaluation for offline learning, and in particular how it would be helpful to have confidence intervals on these estimates such that we could have confident lower bounds. In particular the authors place focus on evaluating the performance of a constantly improving target policy, rather than a fixed checkpoint.

For the OPE method, the authors consider an importance sampling method, a model-based method, and a hybrid approach a la the common weighted doubly-robust method. For the offline RL algorithm, they ablate BC, BCQ, and BEAR.

To get confidence intervals, when doing the off policy evaluation, they do bootstrap resampling of the off-policy estimate $v(\pi_theta)$.

**Audience:**

No

**Claims And Evidence:**

Yes

**Requested Changes:**

I would have appreciated more discussions of when bootstrapping was applied in off-policy evaluation. It's certainly not a new idea (see https://proceedings.mlr.press/v139/hao21b.html for example.)

**Strengths And Weaknesses:**

The authors provide a good introduction to the OPE literature as well as some of the offline RL literature. The application of bootstrapping to get confidence intervals is a quick but reasonable and effective way to get a lower-bound estimate on the value.

I think the primary weaknesses of the paper are:

1) The methods used are a bit far from SOTA for offline RL. Last I checked I believe BEAR and BCQ were considered subpar, in favor of CQL and TD3+BC.

2) The authors claim a contribution of "exploring the effect of offline learning on OPE methods given constantly-improving target policy, as opposed to fixed policy" seems a little hollow. Yes, it is true that many OPE papers present experiment tables for evaluating a fixed policy on a varying number of episodes, but almost every one of those papers in practice is implemented by applying it to a continually learning setup.

3) In general, I found it hard to have a takeaway from the experiment results, aside that bootstrapping led to lower estimates of the offline value function.

I think that the primary contribution of the paper is the application of bootstrapping, except this has been done before in prior work and so I'm not sure this has *any* novel contribution, even a modest one.

---

> ### Author Response · Authors · 2024-05-16
> **Authors' response to reviewer pbpT**
>
> We would like to thank Reviewer pbpT for their detailed review. We find them helpful in improving our work. Below we reply to the requested changes:
> > The methods used are a bit far from SOTA for offline RL. Last I checked I believe BEAR and BCQ were considered subpar, in favor of CQL and TD3+BC.
>
> We understand your point regarding SOTA but our ablation study on different offline RL methods was meant to include methods from different categories of offline algorithms. We agree that there are other algorithms, as of now, that may achieve better performance for offline RL but our goal during the experiments was not mainly to achieve SOTA, but cover a wide range of algorithms (different categories) which will be essential for analyzing the interplay between OPE and offline RL.
>
> > The authors claim a contribution of "exploring the effect of offline learning on OPE methods given constantly-improving target policy, as opposed to fixed policy" seems a little hollow. Yes, it is true that many OPE papers present experiment tables for evaluating a fixed policy on a varying number of episodes, but almost every one of those papers in practice is implemented by applying it to a continually learning setup.
>
> We are not sure we understand the point about OPE papers using a continual learning set-up. But we will try to reply based on our understanding. Yes, policies are often produced by running some RL algorithm but I’m not sure that means it is the same as what we do, as we focus on totally offline RL, and using conservative OPE to evaluate as we learn. The most relevant work to ours in the literature would be: Hyperparameter Selection for Offline Reinforcement Learning [1], which uses OPE for hyperparameter selection in offline RL algorithms and ranking policies. In another work, Benchmarks for Deep off-policy evaluation [2], they select policies collected from snapshots of training a Soft Actor-Critic agent, which covers a range of performance between random and expert. We don’t only propose to use OPE as an evaluation method for offline RL as in [1], or a validation method for tuning parameters as in [2], but we also suggest that is it better to rely on conservative OPE instead as an evaluation method during training (to decide on deployment), to mitigate the overestimation issue and provide further analysis on its performance.
> Please feel free to clarify further if this does not answer your question. Would you like to provide references on what you mean, and we will be happy to clarify the differences.
>
> > In general, I found it hard to have a takeaway from the experiment results, aside that bootstrapping led to lower estimates of the offline value function.
>
> The goal of this work is to co-study the offline RL and OPE literatures, and provide a framework that uses OPE methods jointly with offline RL for conservative evaluation. We have re-structured the analysis section in a way where we provided takeaways in section 5 based on our empirical analysis. These takeaways include recommendations on what OPE method to use, based on different factors such as offline RL algorithm, the environment dynamics and horizon.
>
> > I think that the primary contribution of the paper is the application of bootstrapping, except this has been done before in prior work and so I'm not sure this has any novel contribution, even a modest one.
>
> The primary contribution of this paper is studying the interplay between offline RL and OPE, (which no other work has done similarly); using bootstrapping is one of the essential components we propose to use for evaluating offline RL policies to overcome the the overestimation of OPE.
>
> > I would have appreciated more discussions of when bootstrapping was applied in off-policy evaluation. It's certainly not a new idea (see https://proceedings.mlr.press/v139/hao21b.html for example.)
>
> We are aware of this work, but the paper proposed on bootstrapping is an evaluation paper and does not consider the interplay between offline policy improvement and off-policy evaluation. It is still useful to use as an evaluation method though. Please note that bootstrapping is one of the components in our framework which we use to make evaluation conservative.
>
> Please let us know if any of the points require further clarification.
>
> [1] Hyperparameter Selection for Offline Reinforcement Learning, https://arxiv.org/pdf/2007.09055
>
> [2] Benchmarks for Deep off-policy evaluation, https://arxiv.org/pdf/2103.16596

---

### Review · Reviewer_qFaC · 2024-05-04

**Summary Of Contributions:**

The authors address the question of evaluation of offline policy under the assumption that neither the environment nor the data-generating policy are known.

**Audience:**

No

**Broader Impact Concerns:**

No ethical concerns provided

**Claims And Evidence:**

Yes

**Requested Changes:**

*"In the offline RL literature, there is no standard way to evaluate a policy while
learning offline; most of the literature still evaluates the performance of offline algorithms in the environment
or in a simulator, by running a policy online."*

I think this statement should be put in a wider context. While it is the case that the evaluation of *reported* performance is done in the simulator, numerous works use policy evaluation proxies as a part of the optimisation procedure, including bootstrapping such as in Wang et al (2022) and the cited paper by Hanna et al (2017). There is, however, another related question: can we provide, just based on the offline data, any theoretical guarantees on, for example, the distribution match or any other discrepancy between the offline estimate trajectory and the simulator evaluation? Without such guarantees, it does not answer the original claimed question : "*how to best learn offline and (confidently) select the right time to deploy, if we do not have access to the environment nor the policy generating the data?*"

Further to that, I may just miss the point but it would be good if the authors confirm how the proposed method actually answers 'how to best learn offline'. It appears that the answer to this question is not provided. It is because of these two points that I put the claims and evidence score to 'no' for now.

Also, experimental analysis needs to be improved as detailed above.

Wang, Kerong, et al. "Bootstrapped transformer for offline reinforcement learning." Advances in Neural Information Processing Systems 35 (2022): 34748-34761.

**Strengths And Weaknesses:**

Strengths:
- Evaluation of policies is an important open question, and the paper proposes a sound approach towards solving it.

Weaknesses:
- The method formulation looks not much different from the standard bootstrapping method, so it would be important to emphasise how this algorithm would provide an additional theoretical or empirical value for the readers comparing to existing RL bootstrapping literature, why cannot we gain the same results from, say, Algorithm 1 from Wang et al (2022)? It seems like the same exact results could be achieved with other bootstrapping algorithms for offline RL? If not the authors need to emphasise how exactly this cannot be solved with the existing methods (hence the current score on the audience criterion).
- The experimental evaluation only concerns some simple baselines, which limits the experimental contribution; common gym environments such as halfcheetah, hopper, walker2d could be recommended to improve the experimental evaluation.

---

> ### Author Response · Authors · 2024-05-16
> **Authors' response to reviewer qFaC**
>
> We would like to thank Reviewer qFaC for their review. The points raised are important to discuss. Below we reply to the requested changes:
> > The method formulation looks not much different from the standard bootstrapping method, so it would be important to emphasize how this algorithm would provide an additional theoretical or empirical value for the readers comparing to existing RL bootstrapping literature, why cannot we gain the same results from, say, Algorithm 1 from Wang et al (2022)? It seems like the same exact results could be achieved with other bootstrapping algorithms for offline RL? If not the authors need to emphasise how exactly this cannot be solved with the existing methods (hence the current score on the audience criterion).
>
> In the referenced paper Wang et al (2022), they propose a novel algorithm named Bootstrapped Transformer, which incorporates the idea of bootstrapping and leverages the learned model to self-generate more offline data to further boost the sequence model training. The bootstrapping in our work relies on Efron’s bootstrap [1] to provide lower-bound estimates to the policy values produced by OPE. Their approach is an offline learning algorithm, and bootstrapping in our work is used for conservation evaluation, which is different from what the paper proposed. Bootstrapping is never part of our offline policy optimization; it is only used for evaluation/testing.
>
> > The experimental evaluation only concerns some simple baselines, which limits the experimental contribution; common gym environments such as halfcheetah, hopper, walker2d could be recommended to improve the experimental evaluation.
>
> The OPE literature tends to rely on simpler domains and environments for evaluation, unlike the offline RL literature. We followed the OPE literature as we did 40 runs of each single experiment since OPE methods have high variance.
>
> > I think this statement should be put in a wider context. While it is the case that the evaluation of reported performance is done in the simulator, numerous works use policy evaluation proxies as a part of the optimisation procedure, including bootstrapping such as in Wang et al (2022) and the cited paper by Hanna et al (2017). There is, however, another related question: can we provide, just based on the offline data, any theoretical guarantees on, for example, the distribution match or any other discrepancy between the offline estimate trajectory and the simulator evaluation? Without such guarantees, it does not answer the original claimed question : "how to best learn offline and (confidently) select the right time to deploy, if we do not have access to the environment nor the policy generating the data?"
>
> We build upon the cited paper by Hanna et. al (2017); they use bootstrapping for model-based evaluation, and not part of policy optimization. We follow the same approach as Hanna et. al (2017) in using bootstrapping for evaluation only, not in policy optimization as in Wang et al (2022). While we don’t provide theoretical guarantees to the evaluation algorithm, we provide a workflow for offline policy learning + evaluation. We added insights in the experimental analysis on how offline learning affects evaluation and the choice of OPE method, and other takeaways.
>
> Regarding theoretical guarantees, to avoid duplicate content, we provided detailed reply to reviewer Kxsj's similar concern here: https://openreview.net/forum?id=kLo4TKh0OP&noteId=Qy8F6PPICm
>
> > Further to that, I may just miss the point but it would be good if the authors confirm how the proposed method actually answers 'how to best learn offline'. It appears that the answer to this question is not provided. It is because of these two points that I put the claims and evidence score to 'no' for now.
>
> We understand your concern regarding how to best learn offline. We aimed to explore and study the interplay between offline RL and off-policy evaluation. We have edited the question instead to be: “how to (confidently) select the right time to deploy when learning offline, if we do not have access to the environment nor the policy generating the data?”. In section 5, we added a paragraph highlighting takeaways which answer this question. Briefly, we highlighted that the offline RL algorithm (which controls the divergence between the offline policy and the data’s policy), horizon and environment dynamics are the main factors affecting evaluation. We have also highlighted that it is better to rely on direct methods when it is easy to model the environment dynamics, and rely on importance sampling for shorter horizon environments.
>
> > Also, experimental analysis needs to be improved as detailed above.
>
> We did have more experimental analysis and insights in the appendix. We have rearranged the section in a way that answers the question as you proposed in section 5.
>
> Thanks again, and please let us know if we need to clarify further.
>
> [1] Bootstrap Methods: Another Look at the Jackknife

---

### Author Response · Authors · 2024-05-16
**Summary response to all reviewers with outlined changes**

We would like to thank all reviewers for their time and efforts to provide, which are helping us to make this work better. We would also like to apologize if it took us a long time to reply, given that authors have been at ICLR 2024 with different travel plans.	We have replied below to the requested changes. We have also made the following changes to the paper:
* We have restructured the experimental analysis and included other analysis that we already had in the appendix, to be able to answer the concerns by reviewer qFaC.
* We have added a paragraph (in section 5)  for takeaways to better address the question by reviewer qFaC.
* We have added further discussion on bootstrapping to address reviewer Kxsj.

We would also like to take the chance to clarify our contributions: we propose a framework CEOPL, combining offline RL with conservative off-policy evaluation for both discrete and continuous control tasks. Bootstrapping is only used for our conservative evaluation, since it is not safe to rely on OPE, to produce lower-bound estimates on policy values, but bootstrapping is not used as part of the optimization. We also demonstrate our approach on real-world medical data, which we believe combining offline RL+conservative evaluation will be most useful.

We also reply separately to each reviewer below. Thanks!

---

### Decision · Action_Editor_yujb · 2024-08-23

**Recommendation:** Accept as is

**Comment:**

The reviewers have not reached an argument on this paper, but overall I would lean towards acceptance.

First, one of the negative reviews is justified with a very short "The paper lacks theoretical results.",and I do not think this is enough reason to reject this work. Authors provide sufficient empirical evidence for the usefulness of their framework.

I agree that, from a practitioner's perspective, it does seem new to consider OPE during learning. However, I think we can appreciate that the authors tackle an issue that is strongly relevant to the offline RL community.

**Audience:**

Yes

**Claims And Evidence:**

Yes